# CEP peptide and cytokinin pathways converge on CEPD glutaredoxins to inhibit root growth

Michael Taleski [1,5], Kelly Chapman [1,5], Ondřej Novák [2], Thomas Schmülling [3], Manuel Frank [3,4] ✉ & Michael A. Djordjevic [1] ✉

C-TERMINALLY ENCODED PEPTIDE (CEP) and cytokinin hormones act over short and long distances to control plant responses to environmental cues. CEP and cytokinin pathway mutants share phenotypes, however, it is not known if these pathways intersect. We show that CEP and cytokinin signalling converge on CEP DOWNSTREAM (CEPD) glutaredoxins to inhibit primary root growth. CEP inhibition of root growth was impaired in mutants defective in *trans*-zeatin (*t*Z)-type cytokinin biosynthesis, transport, perception, and output. Concordantly, mutants affected in *CEP RECEPTOR 1* showed reduced root growth inhibition in response to *t*Z, and altered levels of *t*Z-type cytokinins. Grafting and organ-specific hormone treatments showed that *t*Z-mediated root growth inhibition involved *CEPD* activity in roots. By contrast, root growth inhibition by CEP depended on shoot *CEPD* function. The results demonstrate that CEP and cytokinin pathways intersect, and utilise signalling circuits in separate organs involving common glutaredoxin genes to coordinate root growth.

To ensure their survival under diverse conditions, plants evolved complex intercellular communication mechanisms that enable them to sense and respond to environmental cues[1]. To help achieve this, plants have co-opted metabolite- and peptide-hormone pathways to control a myriad of adaptive responses[2–4]. There is a growing number of interactions established between peptide hormones and metabolite hormone pathways. These pathways involve local and systemic signalling to coordinate growth and stress responses throughout the plant[5,6]. One example is the interaction between cytokinin and CLAVATA3/Endosperm surrounding region-related (CLE) hormone pathways which together regulate shoot meristem size[7] and protoxylem vessel formation[8], however little is known about interactions between cytokinin and other peptide hormone pathways.

C-TERMINALLY ENCODED PEPTIDEs (CEPs) play a key role in plant responses to the environment[9]. The *Arabidopsis thaliana* genome has 12 canonical CEP genes that encode propeptide precursors from which one or more 15-amino acid CEP peptide hormones are excised[10,11]. Mature, root-derived CEPs then enter the xylem stream and travel shootward[12–14]. *CEP* gene expression responds to several environmental and nutritional cues including low nitrogen and high carbon[10,15,16]. CEPs interact with CEP RECEPTOR 1 (CEPR1) in *Arabidopsis*[12] and its orthologue COMPACT ROOT ARCHITECTURE 2 (CRA2) in *Medicago truncatula*[17–19]. CEP-CEPR1 signalling inhibits root growth[12,16,18,20], modulates root system architecture and auxin transport[21], and promotes root nodulation[18,22,23], shoot growth[12], and resource allocation for seed yield[24]. In addition, the interaction of CEP1 with CEPR1 promotes nitrate uptake in roots via a systemic

[1]Division of Plant Sciences, Research School of Biology, College of Science, The Australian National University, Canberra, ACT 2601, Australia. [2]Laboratory of Growth Regulators, Faculty of Science, Palacký University & Institute of Experimental Botany, The Czech Academy of Sciences, CZ-783 71 Olomouc, Czech Republic. [3]Institute of Biology/Applied Genetics, Dahlem Centre of Plant Sciences, Freie Universität Berlin, D-14195 Berlin, Germany. [4]Present address: Department of Molecular Biology and Genetics, Aarhus University, 8000 Aarhus, Denmark. [5]These authors contributed equally: Michael Taleski, Kelly Chapman. ✉e-mail: mfrank@mbg.au.dk; michael.djordjevic@anu.edu.au

mechanism[12] by upregulating the genes encoding the phloem-mobile class III glutaredoxins CEP DOWNSTREAM 1 (CEPD1) and CEPD2. CEPD1 and CEPD2 subsequently travel rootward to positively regulate high affinity nitrate transporter gene expression and protein function in the roots[25,26].

Like CEPs, cytokinins also control plant growth in response to the environment and changes in nutritional status[27–30]. ISO-PENTENYLTRANSFERASE 3 (IPT3), IPT5, and IPT7 are major contributors to the biosynthesis of isopentenyladenine (iP)- and trans-zeatin (tZ)-type cytokinins[31]. The P450 enzymes CYP735A1 and CYP735A2 catalyse the conversion of iP-type to tZ-type cytokinins primarily in roots[32,33]. The transporter ATP-BINDING CASSETTE G14 (ABCG14) then enables the shootward translocation of tZ[34,35]. Cytokinins are perceived by ARABIDOPSIS HISTIDINE KINASE (AHK) receptors[36,37]. Double-knockout mutants in AHK2 and AHK3 have increased root system growth and diminished shoot growth[37,38], which resembles the phenotype of cepr1 knockout mutants[12,16]. After AHKs perceive cytokinin, a number of type-B response regulators (ARRs) with redundant functions mediate the transcriptional responses to cytokinin[39,40]. Class III glutaredoxins also appear to play a role in mediating cytokinin-dependent responses, with several acting downstream of type-B ARRs to inhibit root growth in response to high nitrogen[41]. In addition, several class III glutaredoxins also are implicated in plant responses to low nitrogen status[42]. Recently, Ota et al.[43] demonstrated that ABCG14 is necessary for the full upregulation of particular class III glutaredoxin genes under low nitrogen, including CEPD1 and CEPD2. Given that CEPs and cytokinin have similar roles in inhibiting root growth, promoting shoot growth, and in managing whole plant responses to nitrogen status[44–46], it is possible that these pathways intersect.

In this paper, we define a point of intersection of the CEP and cytokinin pathways. Firstly, we used biochemical, genetic and grafting approaches to assess the requirement of cytokinin pathway components for CEP activity and vice versa. We found that mutants affected in tZ synthesis, transport, perception, and signalling showed altered sensitivity to CEP-mediated inhibition of primary root growth and, correspondingly, CEP receptor mutants showed altered sensitivity to tZ. Grafting showed that CEP activity depends on AHK2 and AHK3 in both the root and shoot. In addition, we determined using mass spectrometry that cepr1 mutants had altered cytokinin content, particularly in the level of tZ-type cytokinins. Finally, we showed that tZ and CEP inhibition of primary root growth requires CEPD glutaredoxins. CEPD1 transcript and GFP-CEPD1 levels responded to tZ treatment, and CEPD1 expression was basally altered in ahk2,3. Surprisingly, grafting and organ-specific hormone treatments showed that root growth inhibition by tZ required root CEPD activity, whereas CEPD activity in the shoot was required for CEP responses. Together, these data indicate that CEP and tZ converge on CEPD activity in different organs to inhibit primary root growth. We propose a model where root and shoot signalling circuits involving common glutaredoxin gene targets integrate CEP and tZ pathways to mediate plant responses to environmental and nutritional cues.

## Results and discussion
### CEP inhibition of primary root growth depends on the cytokinin pathway
We first assessed if mutants affected in cytokinin synthesis or transport showed altered sensitivity to CEP3 peptide mediated inhibition of primary root growth[10,20] (Fig. 1; Supplementary Fig. 1). We identified a partial insensitivity to CEP3 in mutants affected in iP and tZ-type

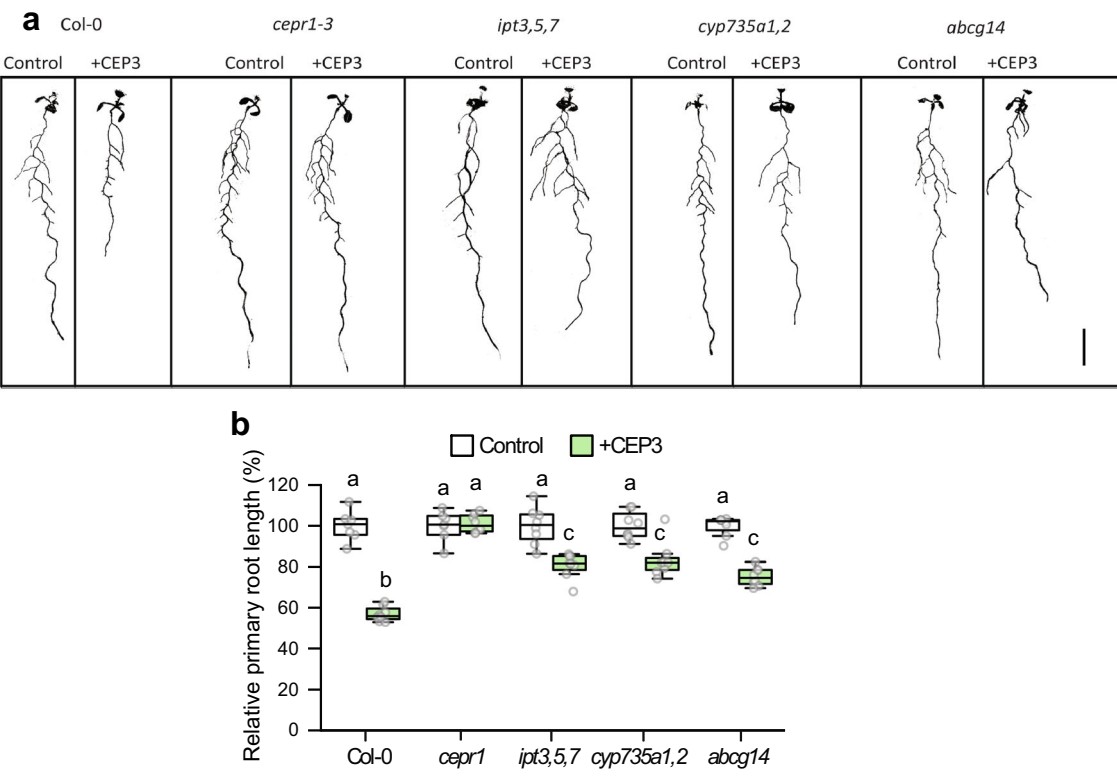

**Fig. 1 | CEP sensitivity depends on cytokinin biosynthesis and transport.**
**a** Representative images and (**b**) relative primary root length for cytokinin biosynthesis or transport mutants grown on medium with or without CEP3 peptide (10⁻⁶ M) for 10 days in comparison to Col-0 and cepr1-3 (n = 7–8 plants). Root length expressed as a percentage of seedlings grown on medium without CEP3 (control) for each respective genotype. Letters show significant differences (ANOVA followed by Tukey HSD test, $p < 0.05$). Scale bar = 1 cm. Box plot centre line, median; box limits, upper and lower quartiles; whiskers, 1.5x interquartile range. See Supplementary Fig. 1 for associated absolute root growth measurements. Exact $p$ values and sample sizes for each treatment group are provided in the Source Data file.

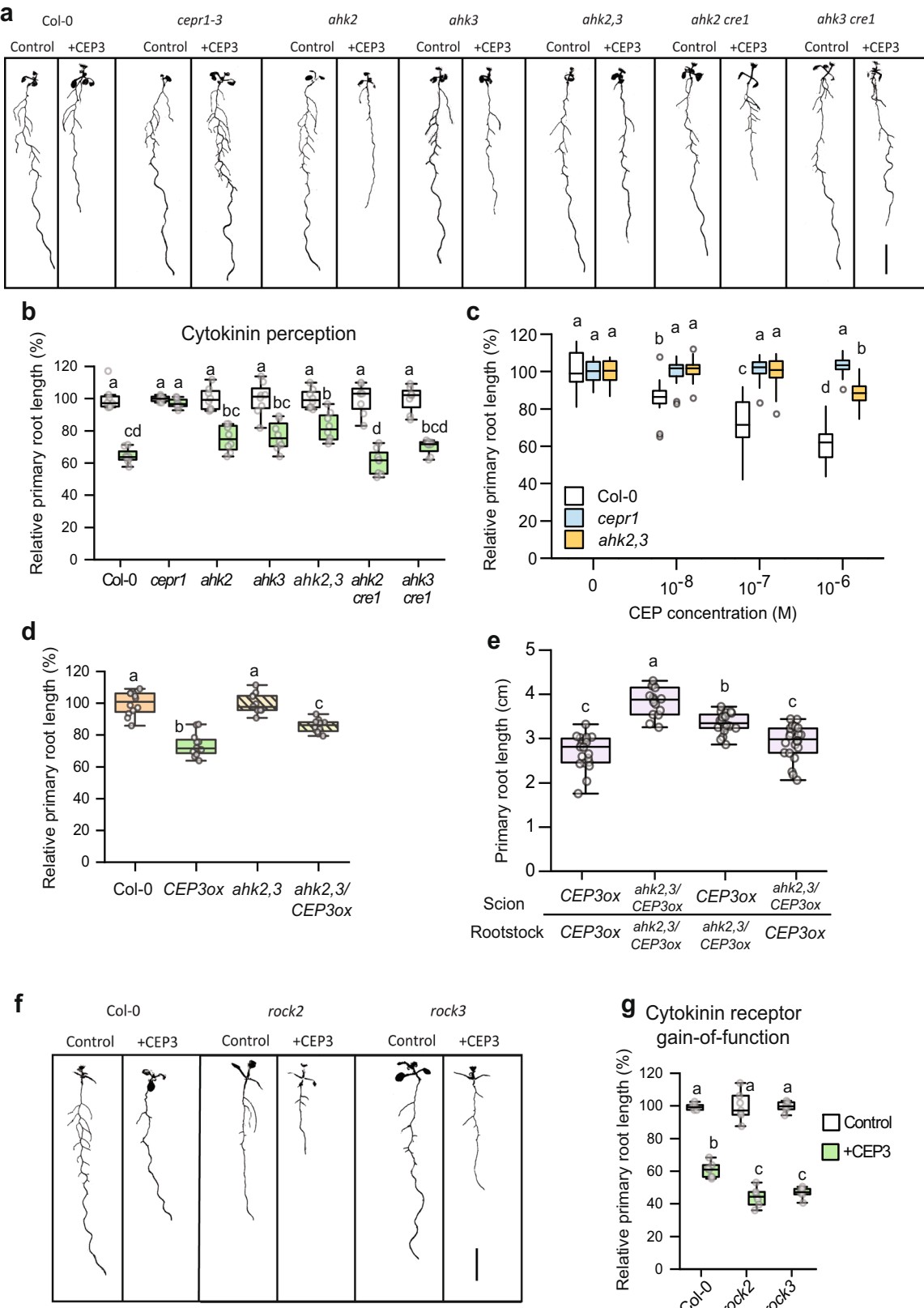

cytokinin biosynthesis (i.e. *ipt3,5,7*), and *t*Z-type synthesis (*cyp735a1*,2) and transport (*abcg14*) (Fig. 1). Next, we examined how cytokinin perception via AHKs contribute to CEP3 sensitivity (Fig. 2; Supplementary Fig. 2). A double knockout mutant affecting AHK2 and AHK3 showed partial insensitivity to CEP3, whereas there was no change in CEP3 sensitivity in the other single or double cytokinin receptor mutants (Fig. 2a,b). To further assess the level of CEP3 insensitivity in *ahk2,3*, we measured relative root growth inhibition to increasing concentrations of CEP3. The *ahk2,3* mutant was circa 100-fold less sensitive to CEP3 than wild type (Fig. 2c).

We then used a CEP3 overexpressing line (*CEP3ox*) crossed into the *ahk2,3* mutant background as an independent strategy to assess

**Fig. 2 | CEP sensitivity depends on cytokinin perception via AHK2 and AHK3 in the root and shoot. a** Representative images and (**b**) relative primary root length for cytokinin perception mutants grown on medium with or without CEP3 peptide ($10^{-6}$ M) for 10 days in comparison to Col-0 and *cepr1-3* (*n* = 8 plants). **c** Sensitivity of *ahk2,3* to increasing CEP3 concentrations (0, $10^{-8}$, $10^{-7}$ or $10^{-6}$ M) at 10 days of growth (*n* = 17-21 plants). Root length expressed as a percentage of seedlings grown on medium without CEP3 for each respective genotype. **d** Effect of *CEP3* over-expression in wild-type and *ahk2,3*. Primary root length after 7 days growth for *CEP3ox* plants was normalised to their respective background control (Col-0 or *ahk2,3*) (*n* = 12 plants). **e** Primary root length for reciprocal hypocotyl grafts

between *CEP3* overexpressing plants in the Col-0 and *ahk2,3* backgrounds (*n* = 15–22 plants). Primary root length measured 12 days post grafting. **f** Representative images and (**g**) relative primary root length of gain-of-function AHK mutants grown on medium with or without CEP3 peptide ($10^{-6}$ M) for 12 days in comparison to Col-0. Letters in (**b-e**, **g**) indicate significant differences (ANOVA followed by Tukey HSD test, $p < 0.05$). Scale bars = 1 cm. Box plot centre line, median; box limits, upper and lower quartiles; whiskers, 1.5x interquartile range. See Supplementary Fig 2 for associated absolute root growth measurements and representative images. Exact *p* values and sample sizes for each treatment group are provided in the Source Data file.

the interaction between CEP signalling and AHK2,3. The *ahk2,3* plants overexpressing *CEP3* showed a smaller reduction in root length than *CEP3*-overexpressing Col-0 (Fig. 2d). To determine if root or shoot *ahk2,3* dampened the effect of *CEP3* overexpression, we reciprocally grafted the *CEP3ox* line (i.e. wild type for *AHK2,3*) with *CEP3* over-expressing *ahk2,3* plants (*ahk2,3/CEP3ox*; Fig. 2e). *CEP3ox* plants with *ahk2,3* in roots only had increased primary root growth relative to *CEP3ox* plants with wild type *AHK2,3* in both roots and shoots, which supported a role for root *AHK2,3* function downstream of CEP3 (Fig. 2e). Maximal root growth, however, was observed in *CEP3ox* plants with *ahk2,3* in both roots and shoots, implying that shoot *AHK2,3* also contributes to root growth inhibition (Fig. 2e). Together, these data support a role for AHK2 and AHK3 activity in both the root and shoot in mediating CEP3-sensitivity for root growth inhibition.

Given that both CEP[20] and cytokinin signalling[47] decrease primary root meristem cell number, we also assessed whether CEP inhibition of meristem cell number depended on AHK2,3 (Supplementary Figs. 3, 4). Meristem cell number in *ahk2,3* was insensitive to CEP3 addition (Supplementary Fig. 3), and partially insensitive to *CEP3* over-expression (Supplementary Fig. 4).

To test further that cytokinin signalling through AHK2,3 plays a specific role in the sensitivity of plants to CEPs, we assessed CEP sen-sitivity in the constitutively-active gain-of-function mutants in either AHK2 (*rock2*) or AHK3 (*rock3*)[48]. Consistently, the *rock2* and *rock3* mutants showed increased CEP sensitivity (Fig. 2f,g). The results observed with the loss- and gain-of-function cytokinin receptor mutants support a role for AHK2 and AHK3 in CEP-mediated inhibition of root growth. Collectively, these results indicate that the full CEP-mediated inhibition of primary root growth involves *tZ* synthesis and transport, and cytokinin perception by AHK2 and AHK3.

We then assessed the requirement for type-B response regulators in mediating CEP inhibition of root growth. We tested the CEP sensi-tivity of two high-order type-B response regulator (*arr*) mutants[39,40]. The *arr1,2,12* mutant was partially insensitive to CEP3, whereas the *arr1,2,10* mutant exhibited no alteration to CEP sensitivity compared to the wild type (Supplementary Fig. 5). These results support a role for particular type B ARRs in CEP-mediated inhibition of root growth.

## CEP and cytokinin signalling pathways intersect

To investigate whether *tZ* inhibition of root growth requires CEP sig-nalling, we assessed the sensitivity of *cepr1* mutants to *tZ* inhibition of primary root growth (Fig. 3; Supplementary Fig. 6). The *cepr1-3* mutant showed a circa 10-fold reduction in sensitivity to *tZ* (Fig. 3a, b). At 10 nM *tZ*, there was a clear differential sensitivity of *cepr1-3* compared to the wild type, and a reduced sensitivity at 100 nM *tZ*.

To determine whether *tZ* and CEP interact, we tested the sen-sitivity of wild type and *cepr1-3* to *tZ* (10 nM), and CEP3 (1 μM) alone or in combination. Whilst *tZ* and CEP3 affected root growth addi-tively in the wild type, *cepr1-3* was insensitive to CEP3 and/or *tZ* treatment (Fig. 3c). This indicates that whilst the effects of CEP3 and cytokinin are additive in the wild type, signalling through CEPR1 affects both *tZ* and CEP-dependent signalling. Collectively, these results indicate an intersection of CEP and cytokinin mediated inhi-bition of primary root growth.

## Cytokinin homoeostasis is perturbed in *cepr1* plants

Mutants affected in cytokinin transport, perception and signalling have perturbed cytokinin content[34,35,37,49]. To examine if CEPR1 affects cytokinin homoeostasis, we measured cytokinin levels in roots and shoots of *cepr1* (Table 1; Supplementary Table 1; Supplementary Data 1). Notably, there were significant perturbations to total root and shoot *tZ*-type cytokinin levels in *cepr1-3* (Col-0) and *cepr1-1* (No-0) compared to wild type (Table 1). Specifically, *tZ*-type cytokinins were elevated in the roots of both *cepr1* mutants. The effects of *cepr1* knockout on *tZ*-type cytokinins in the shoot were ecotype-dependent, with a higher amount in *cepr1-3* and a lower amount in *cepr1-1* com-pared to wild type. In addition, there were some ecotype-dependent perturbations in other cytokinin types in *cepr1* mutants. The altered cytokinin homoeostasis in *cepr1* supports an impairment of cytokinin signalling in these mutants, particularly in *tZ*-type cytokinin responses. The higher amount of *tZ*-type cytokinins in *cepr1* roots is consistent with feedback upregulation of *tZ* biosynthesis resulting from impaired *tZ* signalling in *cepr1* plants, akin to the increased root *tZ* levels observed in *ahk2,3* mutants[37,50].

## *CEPD1* expression is induced by *tZ* and affected by AHK2,3 activity

CEP- and *tZ*-dependent pathways intersect to inhibit root growth, however, it is unknown how the two pathways converge. Recently, Ota et al.[43] showed that low nitrogen-dependent upregulation of the glutaredoxin-encoding genes *CEPD1* and *CEPD2* involved the cytokinin transporter ABCG14. Therefore, we evaluated whether *CEPD* expres-sion depended on cytokinin signalling (Fig. 4). Firstly, we determined if *CEPD1* and *CEPD2* expression in roots or shoots was regulated by *tZ* treatment (Fig. 4a–c). *tZ* treatment upregulated the cytokinin marker gene *ARR5* in both roots and shoots at 8 h post treatment, confirming that a cytokinin response was elicited (Fig. 4a). Simultaneously, *CEPD1* was significantly upregulated in roots by *tZ* (Fig. 4b), whereas *CEPD2* expression was not significantly altered (Fig. 4c). Moreover, GFP-CEPD1 fluorescence was increased in the primary root vasculature region in response to *tZ* or CEP3 treatment (Fig. 4d, e). In addition, *CEPD1* transcript levels were basally lower in the *ahk2,3* background compared to wild type but *CEPD1* transcription remained respon-sive to CEP3 treatment (Fig. 4f). Additionally, we tested if CEP3 peptide treatment upregulates *CEPDs* in the shoot, as has been shown for the CEP1 peptide[25]. CEP3 upregulated *CEPD1* and *CEPD2* transcripts in shoots (Fig. 4g). Moreover, CEP3 treatment, but not *tZ* treatment, upregulated GFP-CEPD1 fluorescence in the cotyledon vasculature (Fig. 4h, i). These data suggest that *CEPD1* transcript levels in roots are induced by *tZ* and are affected by AHK2,3 activity, and that CEP3 can increase *CEPD* expression in both roots and shoots.

## *tZ* sensitivity is affected by root *CEPD* activity

Given that *CEPD* expression is affected by cytokinin signalling, we assessed whether CEPDs are functionally involved in root growth inhibition in response to *tZ* (Fig. 5; Supplementary Fig. 7). The *cepd1,2* double mutant (No-0 ecotype) was partially insensitive to *tZ* applica-tion with respect to inhibition of primary root growth (Fig. 5a, b) and meristem cell number (Supplementary Fig. 8). As CEPDs are phloem-

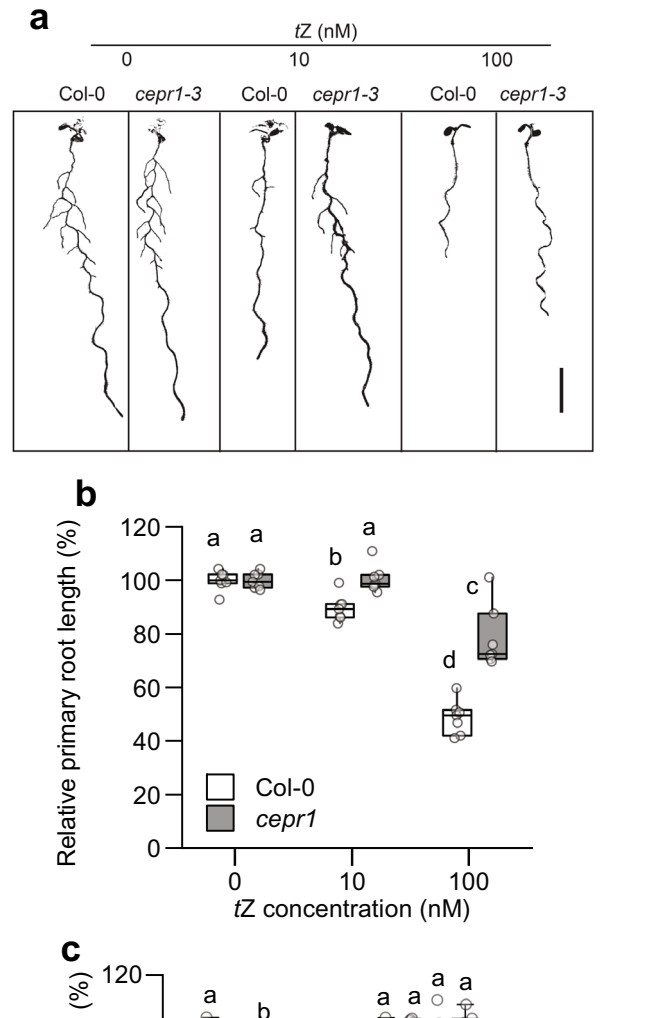

Fig. 3 | CEP and cytokinin signalling pathways intersect. a Representative images and b relative primary root length for Col-0 and cepr1-3 in response to increasing concentrations of tZ (0, 10, 100 nM) at 12 dg. Root length expressed as a percentage of seedlings grown on medium without tZ for each respective genotype (n = 7 plants). c CEP and tZ signalling is defective in cepr1-3. Primary root length was measured for Col-0 and cepr1-3 plants grown on medium containing CEP3 (1 μM), tZ (10 nM), combined CEP3 and tZ, or control medium (no CEP or tZ) for 12 days (n = 7 plants). Root length expressed as a percentage of the control plants. Letters represent significant differences (ANOVA followed by Tukey HSD test, p < 0.05). Scale bar = 1 cm. Box plot centre line, median; box limits, upper and lower quartiles; whiskers, 1.5x interquartile range. See Supplementary Fig. 6 for associated absolute root growth measurements and representative images. Exact p values are provided in the Source Data file.

mobile, shoot-to-root signals[25], we determined via reciprocal hypocotyl grafting if root or shoot CEPD activity was required for root growth inhibition by tZ. Surprisingly, having a cepd1,2 root genotype was sufficient to impart a reduced sensitivity to tZ (Fig. 5c). To further test

**Table 1 | Total cytokinin content in roots and shoots of wild type and cepr1 plants**

| Line | Tissue | tZ-types | iP-types | cZ-types | DHZ-types |
|---|---|---|---|---|---|
| Col-0 | Roots | 9.57 ± 0.92 | 11.38 ± 0.61 | 35.57 ± 2.07 | 1.66 ± 0.18 |
| cepr1-3 | | 16.44 ± 2.50*** | 11.68 ± 0.85 | 32.52 ± 2.79 | 2.55 ± 0.39 |
| Col-0 | Shoots | 13.91 ± 0.80 | 39.52 ± 3.08 | 25.80 ± 2.08 | 1.67 ± 0.07 |
| cepr1-3 | | 20.39 ± 1.45*** | 34.06 ± 8.00 | 26.54 ± 2.10 | 2.65 ± 0.24** |
| No-0 | Roots | 10.23 ± 0.69 | 23.34 ± 1.24 | 63.51 ± 3.42 | 0.68 ± 0.06 |
| cepr1-1 | | 14.69 ± 0.99*** | 23.91 ± 2.00 | 59.05 ± 5.05 | 0.88 ± 0.16* |
| No-0 | Shoots | 13.64 ± 1.19 | 49.51 ± 1.32 | 28.18 ± 1.38 | 1.93 ± 0.25 |
| cepr1-1 | | 10.50 ± 1.39** | 44.71 ± 2.58* | 32.12 ± 1.38** | 1.90 ± 0.10 |

Cytokinin content was measured in root and shoot samples from 6-d-old seedlings grown on standard ½ MS medium. Values are in pmol g⁻¹ fresh weight. Data are mean ± S.D. (n = 5 biologically independent samples). Asterisks indicate statistically significant differences from wild type (two-sided two sample t test without adjustments for multiple comparisons; *p < 0.05, **p < 0.01, ***p < 0.01). See Supplementary Table 1 for levels of individual cytokinin metabolites. Exact p values are provided in Supplementary Data 1.
tZ trans-zeatin, iP isopentenyladenine, cZ cis-zeatin, DHZ dihydrozeatin.

if local or long-distance signalling was involved in tZ-dependent root growth inhibition, we utilised a segmented agar plate set-up to specifically treat roots or shoots with tZ (Fig. 5d). tZ treatment of roots was sufficient for root growth inhibition, and the extent of root growth inhibition was reduced in cepd1,2 (Fig. 5d). Therefore, in addition to its known role as a shoot-derived signal in the CEP pathway, cytokinin-mediated inhibition of root growth depends on a local signalling circuit involving root-derived CEPD.

**Shoot CEPD activity is required for CEP inhibition of primary root growth**

Whilst shoot CEPD function is important for CEP signalling[25], a role for root CEPDs is plausible given root CEPD1 transcripts are upregulated by CEP (Fig. 4f) and are downregulated in Arabidopsis cepr1[16], and Medicago cra2 lines[23]. Therefore, we investigated if CEPDs were involved in CEP-dependent root growth inhibition and whether this required root or shoot CEPD function (Fig. 6, Supplementary Fig. 9). The cepd1,2 double mutant was partially insensitive to CEP3 application (Fig. 6a, b). Reciprocal hypocotyl grafting demonstrated that shoot cepd1,2 was sufficient to diminish sensitivity to CEP3 (Fig. 6c). We next confirmed using a segmented agar plate set-up that shoot-applied CEP3 was sufficient to impart maximal root growth inhibition (Supplementary Fig. 9d, e), and that cepd1,2 was less sensitive to shoot applied CEP3 (Fig. 6d). These results demonstrate that the CEP and cytokinin pathways converge on CEPD activity in different organs to modulate root growth.

**Environmental cues trigger cytokinin and CEP pathways that converge on CEPDs to control root growth**

CEPs and cytokinins control related processes in plants including root growth[10,16,20,37,51], root nodule formation[18,52], nutrient mobilisation and utilisation[16,24,53], auxin transport[21,54] and yield[24,55]. Until now, definitive links or overlaps between cytokinin and CEP signalling remained obscure or had not been functionally verified.

From the results here and from prior work[10,12,15,16,23,25,43,45], we propose the following model where CEP and tZ pathways converge on CEPD glutaredoxins (Fig. 7). Environmental or internal (e.g. nutrient status) stimuli trigger the production of CEP and tZ-type cytokinin signals in roots. These can act locally in roots, or enter the xylem stream for translocation to the shoot. CEP perception by CEPR1 inhibits root growth via shoot-derived CEPDs, whereas tZ perception by AHK2,3 inhibits root growth locally via root-derived CEPDs. Crosstalk occurs between the CEP and tZ pathways as (i) CEP-CEPR1 signalling results in feedback inhibition of tZ levels in roots, (ii) CEP-CEPR1 regulates root CEPD expression[16,23], thus affecting tZ-sensitivity, and (iii) tZ

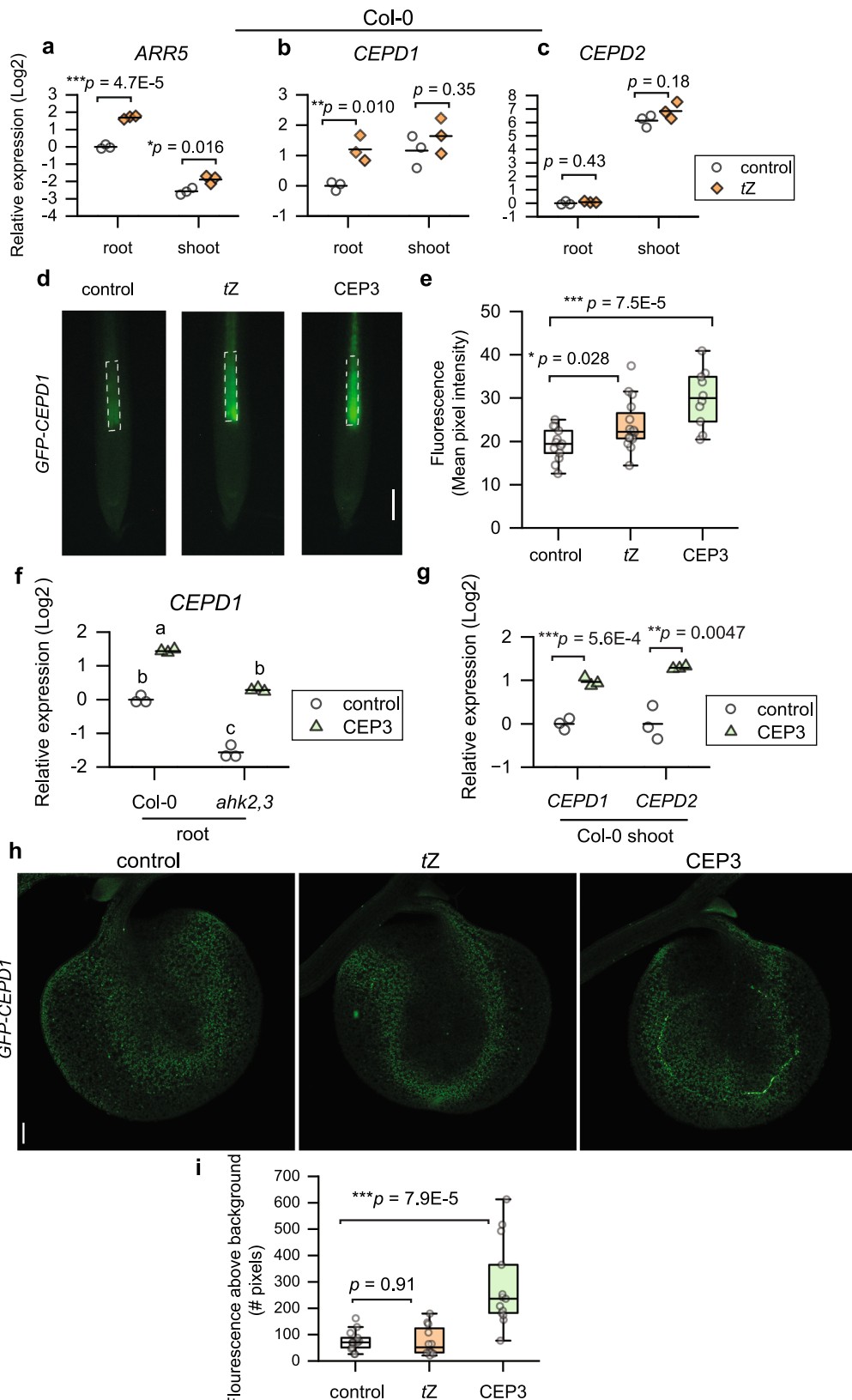

transport to the shoot impacts shoot *CEPD* expression under low nitrogen[43], affecting CEP-sensitivity. CEPDs therefore act as a convergence point to integrate CEP and *tZ*-type cytokinin signalling to fine tune root growth in response to a spectrum of stimuli. Given that *cepd1,2* mutants are not fully insensitive to CEP or *tZ* treatment, this suggests other molecular components also contribute to CEP and

cytokinin downstream signalling. It is plausible that additional type III glutaredoxin members[41,43], or other yet to be determined signalling components, contribute to CEP/cytokinin signalling and crosstalk in addition to CEPDs. Establishing the generalisability of CEPD's contribution to CEP and cytokinin signalling will also require future studies utilising *cepd* mutants in different Arabidopsis ecotypes as well as in

**Fig. 4 | Root *CEPD1* expression is induced by *tZ* and is dependent on AHK2,3 activity. a** *ARR5*, (**b**) *CEPD1*, and (**c**) *CEPD2* expression in roots and shoots in response to *tZ* treatment (8 h). *n* = 3 biologically independent samples containing ~50 roots, or ~30 shoots. **d** Representative images and (**e**) fluorescence intensity of GFP-CEPD1 in response to *tZ* or CEP3 treatment (24 h) in the primary root vasculature region (white dashed box). *n* = 10–16 plants. **f** *CEPD1* expression in roots responds to CEP treatment (8 h) and is basally reduced in *ahk2,3*. **g** *CEPD1* and *CEPD2* expression in the shoots respond to CEP3 treatment (8 h). *n* = 3 biologically independent samples containing ~50 roots, or ~20 shoots. **h** Representative images and **i** GFP-CEPD1 fluorescence in the cotyledon above background (pixels with intensity value above 26,000) in response to *tZ* or CEP3 treatment (24 h). *n* = 12-14 cotyledons. Five (**d**, **e**, **h**, **i**) or 6 day old (**a**–**c**, **f**, **g**) seedlings were transferred for specified durations to medium with DMSO (control), *tZ* (10 nM), or CEP3 (1 μM). Significant differences determined by a two-sided two-sample *t* test without adjustments for multiple comparisons; *$p < 0.05$, **$p < 0.01$, ***$p < 0.001$ (**a**–**c**, **e**, **g**, **i**) or ANOVA followed by Tukey HSD test, $p < 0.05$ (**f**). Exact *p* values for (**f**) are provided in the Source Data file. Box plot centre line, median; box limits, upper and lower quartiles; whiskers, 1.5x interquartile range. Scale bars: (**d**) = 100 μm; (**i**) = 200 μm. Exact sample sizes for each treatment group for (**d**) and (**h**) are provided in the Source Data file.

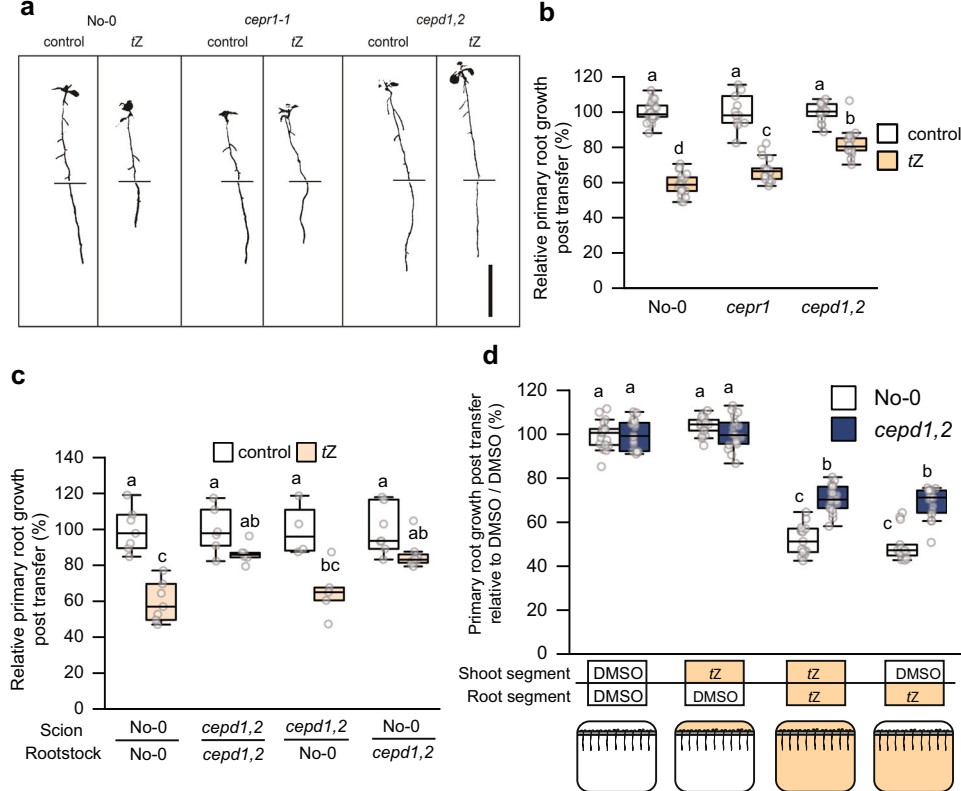

**Fig. 5 | Cytokinin signalling through root *CEPDs* inhibits primary root growth. a** Representative images and (**b**) relative primary root growth for No-0, *cepr1-1*, and the *cepd1,2* double mutant seedlings 3 days after transfer to medium with or without *tZ* (10 nM) (n = 11–24 plants). Horizontal bars in (**a**) indicate the length of the primary root on the day of transfer (i.e. 6 days growth). Root growth post transfer expressed as a percentage of seedlings grown on solvent control (DMSO) for each respective genotype. **c** *tZ* sensitivity of reciprocal hypocotyl grafts between No-0 and *cepd1,2*. Primary root growth 3 days post transfer to *tZ* (5 nM) expressed as a percentage of seedlings transferred to DMSO (control) for each respective graft combination (*n* = 4–8 plants). **d** Relative primary root growth for No-0 and *cepd1,2* seedlings 3 days post transfer to segmented plates for *tZ* treatment of roots and/or shoots. See lower panel for diagrammatic representation of the segmented agar plate set-up. Plants were grown for 6 days before transfer to segmented plates with *tZ* (10 nM) or solvent control (DMSO) infused in each plate segment. Plants were positioned such that roots only were in contact with the bottom segment, and shoots only were in contact with the top segment. Root growth post transfer expressed as a percentage of the DMSO control for each respective genotype (*n* = 13–18 plants). Letters in (**b**–**d**) show significant differences (ANOVA followed by Tukey HSD test, $p < 0.05$). Scale bar = 1 cm. Box plot centre line, median; box limits, upper and lower quartiles; whiskers, 1.5x interquartile range. See Supplementary Fig 7 for associated absolute root growth measurements and representative images. Exact *p* values and sample sizes for each treatment group are provided in the Source Data file.

different plant species. Whether hormone crosstalk in control of root growth also involves cytokinin-dependent regulation of *CEP* gene expression, as recently shown in legume nodulation[56], remains an open question. From a nitrogen centric perspective, the convergence of low nitrogen (i.e. CEP)- and high nitrogen (i.e. cytokinin)- induced hormone pathways on class III glutaredoxin activity implies that plants utilise common gene targets to slow root growth under disparate conditions such as under nitrate starvation or excess. More broadly, as both *CEP* and cytokinin are induced by elevated sugar[16,57], it is possible that these pathways converge on CEPD glutaredoxin function to balance root growth and nitrogen acquisition with the availability of photosynthetically-derived carbon.

## Methods

### Plant materials and growth conditions

In the *Arabidopsis thaliana* Col-0 ecotype, the *cepr1-3* (467C01; GABI-Kat)[16,58], *CEP3ox*[10], *ahk2-5*, *ahk3-7*, *cre1-2*, *ahk2-5 ahk3-7*, *ahk2-5 cre1-2* and *ahk3-7 cre1-2*[37], *abcg14-2*[34,35], *ipt3-2 ipt5-2 ipt7-1*[31], *cyp735a1-2 cyp735a2-2*[33], *rock2-1* and *rock3-1*[48] lines were used. The *arr* triple mutants (*arr1,2,10* and *arr1,2,12*; Col-0) were generated by crossing *arr1-3* and *arr10-5* or *arr12-1*[39], with *arr2* (GK-269G01)[59]. The *ahk2,3/CEP3ox* line was generated

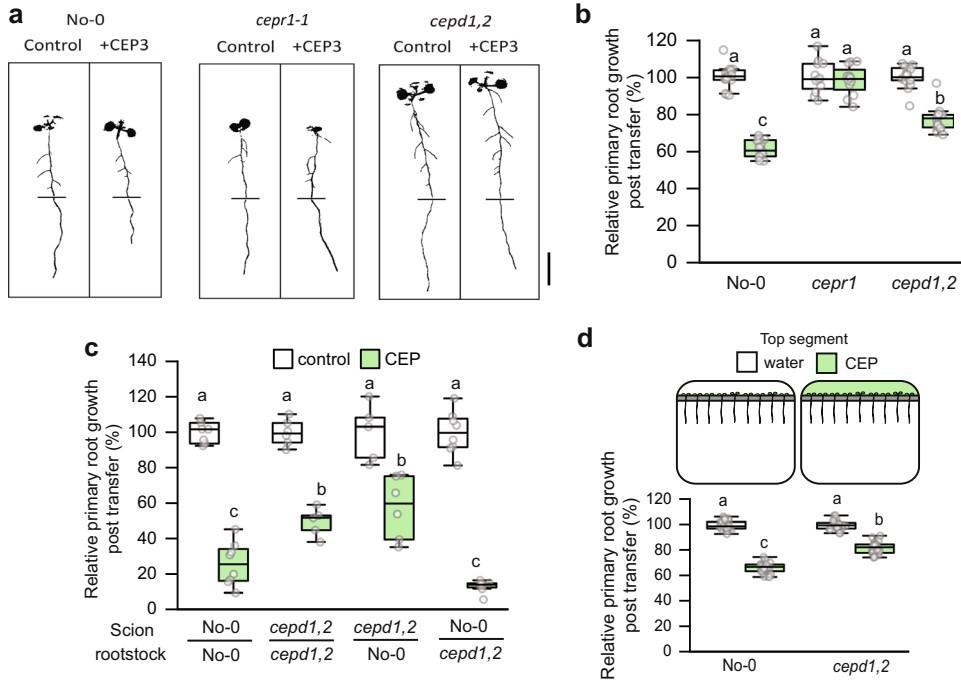

**Fig. 6 | CEP signalling through shoot *CEPDs* inhibits primary root growth.**
**a** Representative images and (**b**) relative primary root growth for No-0, *cepr1-1*, and the *cepd1,2* double mutant seedlings 3 days after transfer to medium with or without CEP3 ($10^{-6}$ M) (*n* = 12–18 plants). Horizontal bars in (**a**) indicate the length of the primary root on the day of transfer (i.e. 6 days growth). Root growth post transfer expressed as a percentage of seedlings grown on medium without CEP3 (control) for each respective genotype. **c** CEP3 sensitivity of reciprocal hypocotyl grafts between No-0 and *cepd1,2*. Root growth 3 days post transfer to treatment plates expressed as a percentage of seedlings grown on medium without CEP3 (control) for each respective graft combination (*n* = 5–8 plants). **d** A diagrammatic representation (upper) and relative primary root growth (lower) for No-0 and

*cepd1,2* seedlings 3 days post transfer to segmented plates for selective CEP3 treatment of shoots. Plants were grown for 6 days before transfer to segmented plates with CEP3 ($10^{-6}$ M) or solvent control (water) infused in the top agar segment. Plants were positioned such that shoots but not roots were in contact with the top segment. Root growth post transfer expressed as a percentage of the water control for each respective genotype (*n* = 22–23 plants). Letters in (**b**–**d**) show significant differences (ANOVA followed by Tukey HSD test, *p* < 0.05). Scale bar = 1 cm. Box plot centre line, median; box limits, upper and lower quartiles; whiskers, 1.5x interquartile range. See Supplementary Fig 9 for associated absolute root growth measurements, representative images, and extended data. Exact *p* values and sample sizes for each treatment group are provided in the Source Data file.

---

by crossing *CEP3ox*[10] and *ahk2-5 ahk3-7*[37], and F3 or F4 progeny homozygous for all transgenes were used. In the Nossen (No-0) ecotype, the *cepr1-1* (RATM11-2459; RIKEN)[12,60] *cepd1-1 cepd2-1*[25], and GFP-CEPD1 (in *cepd1-1 cepd2-1*)[25] lines were used. Sterilised seeds were grown on solidified media (1% Type M agar) containing ½ strength Murashige–Skoog (MS) basal salts (Sigma) at pH 5.7. Medium was supplemented with 1% w/v sucrose for experiments in Figs. 1–3, and Supplementary Figs. 1, 2 and 4. Medium for all other experiments contained no added sucrose, unless otherwise indicated. Plates were grown in chambers at 22 °C with a 16 h photoperiod with 100–120 μmol m$^{-2}$ s$^{-1}$ light. Roots were scanned on a flatbed scanner and primary root lengths were measured using ImageJ with the SmartRoot plugin[61].

## CEP peptide and *trans*-zeatin treatments
Synthetic AtCEP3 (TFRhyPTEPGHShyPGIGH) was dissolved in $H_2O$[10]. CEP3 peptide was synthesised by GL Biochem, Shanghai, and the structure was validated independently by mass spectrometry. *trans*-zeatin (Sigma) was dissolved in DMSO. CEP3 and *trans*-zeatin were added to media after autoclaving and used at the concentrations indicated.

## Grafting
Grafting was performed as described previously[21]. Briefly, seedlings were grown for 6 days on ½ MS with 0.5% sucrose prior to hypocotyl grafting. Both cotyledons were removed then scions were cut and transplanted to the new rootstock hypocotyl. Grafts recovered for five days before subsequent plate transfers. For Fig. 2e, grafted plants were transferred to ½ MS medium with 1% sucrose and grown for a further

7 days before primary root length was measured. For Fig. 5c and Fig. 6c, plants were first transferred to ½ MS medium (no sucrose) for 3 days, and then to ½ MS medium (no sucrose) treatment plates with or without *tZ* (5 nM) or CEP3 (1 μM), respectively. Plants were grown for a further 3 days on treatment plates before primary root growth post transfer was measured.

## Microscopy
Root analyses were conducted with a Leica DM5500 microscope. For enumeration of cortical cells in the meristematic zone, primary root tips were stained with propidium iodide (100 μM, 2 min) and imaged using excitation at 540-580 nm and emission at 592-668 nm[16,20]. For quantification of GFP-CEPD1 fluorescence, the primary root vasculature region was imaged using excitation at 490-510 nm and emission at 520–550 nm. Mean pixel intensity was determined using ImageJ (http://rsb.info.nih.gov/ij/). Cotyledon analyses were conducted with a Zeiss LSM800 with Airyscan confocal microscope (Centre for Advanced Microscopy, ANU) using excitation at 488 nm, emission at 509 nm and detection at 410-550 nm. Pixels in the region corresponding to the cotyledon with intensity values above 26000 were counted using ImageJ, as this threshold minimised the presence of background fluorescence outside the cotyledon vasculature.

## Quantification of endogenous cytokinins
The concentrations of endogenous cytokinins were determined in roots and shoots of 6-d-old *Arabidopsis* plants grown on ½ MS plates. For each sample, pooled roots or shoots (7 to 25 mg fresh weight) were snap frozen in liquid nitrogen and subsequently freeze-dried. Five

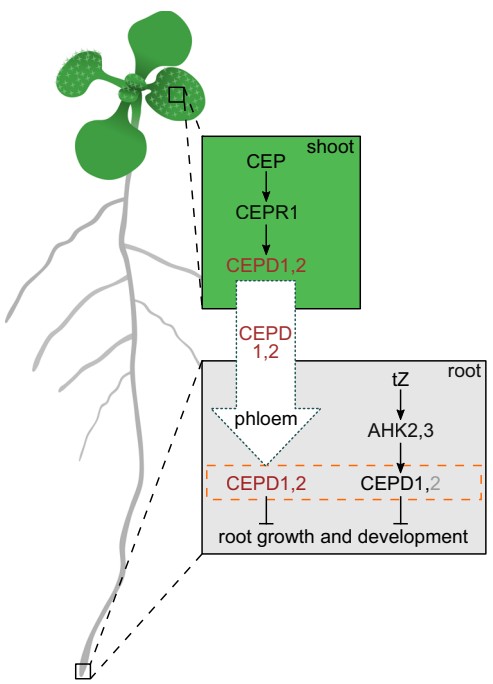

**Fig. 7 | CEP peptide and cytokinin pathways converge on CEPD glutaredoxins to inhibit root growth.** In the shoot, the perception of root-derived CEP by CEPR1 results in increased CEPD1,2 glutaredoxin production (red text). Shoot derived CEPD1,2 travels to the root via the phloem. In the root, *tZ* signals locally through AHK2,3 to transcriptionally induce local CEPD1 production (black text). It is possible that *tZ* signals through CEPD2 (grey text) locally via other mechanisms (e.g. post-translationally). The pool of shoot- and root-derived CEPD (orange box with broken lines) ultimately integrates root growth responses with whole-of-plant nutritional and environmental stress status. *tZ trans*-zeatin, AHK ARABIDOPSIS HISTIDINE KINASE, CEP C-TERMINALLY ENCODED PEPTIDE, CEPR CEP RECEPTOR, CEPD CEP DOWNSTREAM.

independent biological replicates were analysed for each genotype and tissue. Sample extraction and purification were performed according to the method described previously[62], with modifications[63]. Briefly, samples (approx. 20 mg fresh weight) were homogenized and extracted in 1 mL of modified Bieleski buffer (60% MeOH, 10% HCOOH and 30% $H_2O$) together with a cocktail of stable isotope-labelled internal standards (0.2 pmol of cytokinin bases, ribosides, *N*-gluco-sides, and 0.5 pmol of cytokinin *O*-glucosides, nucleotides per sample added). The extracts were applied onto an Oasis MCX column (30 mg/1 mL, Waters) conditioned with 1 mL each of 100% MeOH and $H_2O$, equilibrated sequentially with 1 mL of 50% (v/v) nitric acid, 1 mL of $H_2O$, and 1 mL of 1 M HCOOH, and washed with 1 mL of 1 M HCOOH and 1 mL 100% MeOH. Analytes were then eluted by two-step elution using 1 mL of 0.35 M $NH_4OH$ aqueous solution and 2 ml of 0.35 M $NH_4OH$ in 60% (v/v) MeOH solution. The eluates were then evaporated to dryness in vacuo and stored at −20 °C. Quantification of endogenous cytokinins was done by ultra-high performance liquid chromatography–fast scanning tandem mass spectrometry, using stable isotope-labelled cytokinin standards[62]. Separation was performed on an Acquity UPLC® i-Class System (Waters, Milford, MA, USA) equipped with an Acquity UPLC BEH Shield RP18 column (150 × 2.1 mm, 1.7 μm; Waters), and the effluent was introduced into the electrospray ion source of a triple quadrupole mass spectrometer Xevo™ TQ-S MS (Waters), operating in multiple reaction monitoring (MRM) mode. Cytokinin concentrations were determined using MassLynx software (version 4.2; Waters) using stable isotope dilution method (Supplementary Data 1). Five independent biological replicates were performed, including two technical replicates of each.

## qRT-PCR

Total RNA was isolated from harvested tissue snap frozen in liquid nitrogen using a modified Trizol extraction method using columns from the RNeasy plant mini kit (QIAGEN)[10]. cDNA synthesis was performed using oligo(dT)12–18 primers and Superscript III reverse transcriptase (Invitrogen). qRT-PCR was conducted using Fast SYBR Green fluorescent dye (Applied Biosystems) and samples were run on a ViiA 7 Real-Time PCR System (Applied Biosystems) according to manufacturer's specifications. Data were analysed using the ΔΔCT method[64], with *EF1α* (At1g07920) expression used for normalisation[65]. Primers used are listed in Supplementary Table 2.

## Segmented agar plate assays

For segmented agar plates, a 5 mm trench in the agar was excavated 18 mm from the top of the plate. Solutions of CEP, *tZ*, or solvent controls were applied to final specified concentrations, spread evenly across the root or shoot agar segment, and allowed to diffuse overnight. After 6 days of growth on standard agar plates, plants were transferred to segmented plates and positioned such that roots only were in contact with the bottom segment, and shoots only were in contact with the top segment.

## Accession numbers

Arabidopsis Genome Initiative locus codes are as follows: *CEPR1*, AT5G49660; *CEPD1*, AT1G06830; *CEPD2*, AT2G47880; *AHK2*, AT5G35750; *AHK3*, AT1G27320.

## Reporting summary

Further information on research design is available in the Nature Portfolio Reporting Summary linked to this article.

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

## Acknowledgements

This work was supported by an Australian Research Council grant to MAD (DP200101885) and by the Ministry of Education, Youth and Sports of the Czech Republic (European Regional Development Fund-Project "Plants as a tool for sustainable global development" No. CZ.02.1.01/0.0/0.0/16_019/0000827). We thank Bernd Weisshaar (MPI for Plant Breeding Research, Cologne) for the T-DNA mutant line 467C01, and RIKEN (Japan) for providing the RATM11-2459, *cepd1-1 cepd2-1* and GFP-CEPD1 lines. The authors acknowledge the instruments and expertise of Microscopy Australia at the Centre for Advanced Microscopy, Australian National University, a facility that is funded by the University and the Federal Government through NCRIS. We thank Sören Werner and Gabi Grüschow for generating the *arr* mutants. The authors give sincere thanks to Hana Martínková and Ivan Petřík for their help with cytokinin analyses.

## Author contributions

M.F., M.A.D., M.T. and K.C. conceived and designed experiments; T.S. provided plant materials; K.C., M.T. and M.F. performed experiments; O.N. conducted the cytokinin quantification; M.T., M.F., M.A.D and K.C. analysed the data. M.A.D., M.T., K.C. and M.F. wrote the manuscript; T.S. and O.N. participated in drafting.

## Competing interests

The authors declare no competing interests.
