## [Peer Review File · Nature Communications]

CEP peptide and cytokinin pathways converge on CEPD glutaredoxins to inhibit root growthReviewer #1 (Remarks to the Author):

In the present manuscript, Taleski and collaborators investigate possible interactions between CEP3, C-terminally encoded peptide 3, and cytokinins (CK), which are both known to be part of nutritional systemic signaling pathways. To do so, the authors used a combination of genetic, grafting and "pharmacological" (CEP3 or tZ external supply) approaches.

Results and discussion are combined and divided in 6 parts summarized below:

- 1) The role of CK in the CEP3-dependent inhibition of the primary root growth is assessed using a set of CK biosynthesis (*ipt3,5,7*, *cyp735a1,2*), transport (*abcg14*) and signaling mutants (*ahks*, type-B *arrs*). In *ipt3,5,7*, *cyp735a1,2*, *abcg14*, *ahk2,3* (mutant and gain-of-function) and *arr1,2,12* mutants, the inhibition of the primary root growth by CEP3 is altered (i.e., reduced in mutants).
- 2) Conversely, the inhibition of primary root growth by exogenous tZ application would require the CEP receptor as indicated by a less/absence of tZ-response in *cepr1* mutant.
- 3) Grafting experiments using a CEP3 overexpressor line and *ahk2,3* mutant show that i) CEP3 expression in roots has the greatest effect on primary root growth and ii) that this inhibition is reduced when AHK2 and 3 are mutated in the shoots
- 4) CK content analysis showed that the concentration of tZ-types is affected in roots and shoots by *cepr1* mutation (higher tZ-type content in root).
- 5) CEPD1 and 2, belonging to glutaredoxin family, are downstream CEP/CEPR to regulate NRT2.1 expression in response to N-limitation. Here, the authors showed that CEPD1 and 2 are required for CEP and tZ- inhibition of primary root growth.
- 6) This last paragraph proposes a model of interaction between CEP and tZ in response to environmental cues.

These results are original and interesting since i) CEP/CEPR/CEPD signaling pathway was known to regulate NRT2.1 and not known to regulate the primary root growth, although CEP3/CEPR1 was known to be involved (Delay et al. 2019), ii) the interaction between CEPDL (CEPD-like) and tZ has been shown but not between CEPD signaling pathway and tZ.

To my opinion, these results are interesting but may be presented differently to highlight their novelty. The role of CEPD1&2 in CEP3-dependent primary root growth regulation has not been shown previously. However, this result is presented only in the middle of the graph in Figure 4. Why the authors did not show this result in a 1st figure? In addition, I think that the study would have a greater impact if the authors determine whether or not the control of the primary root growth by CEPDs depends on their activity in shoots or roots (grafting)? Indeed, previously the authors have shown that the control of the primary root growth by CEP3 depends on *cepr1* activity in roots and shoots. So, this experiment would clarify the role of this signaling pathway at a systemic level.

Figures 1 and 2 display very clear results on the role of CK in CEP3-mediated root growth inhibition and CEP/CEPR in tZ-mediated root growth inhibition. Therefore, I would suggest to add in Figure 2 the results presented in Figure 4 (only the +tZ treatment in *cepr1* and *cepd1/2* mutants).

In Figure 3, the role of *ahk2,3* in CEP3-mediated primary root growth inhibition seems to depend on the activity of these receptors in shoots. However, I think that a last control is missing to reinforce the conclusion. Indeed, CEP3ox has the strongest phenotype when used as the rootstock but the role of *ahk2,3* in roots is tested only with CEP3ox shoots. Therefore, I would recommend to obtain a *ahk2,3* mutant overexpressing CEP3 and to use it as a rootstock (this control could even replace the last bar of the graph Figure 3b). To comfort the model, in particular box2, it will be worth to add a similar study using *arr1/2/12* mutants (Additional remark: what is the rationale to focus only on these ARR's? I guess it's their link with WUSCHEL activation?).

The impact of this study will be also reinforced if the authors would add some grafting experiments combining *cepr1* mutant and +tZ treatments, in order to parallel the study

with the CK receptor.

In Table 1, the authors showed that the accumulation of tZ is altered in *cepr1* mutants, which is a very interesting result as well. Unfortunately, this result is only described in the corresponding paragraph but not really interpreted and not even displayed in the model (Figure 5). The results show most of the time that tZ concentration is higher in mutants whereas all the other concentrations are not modified. What does it mean? Is the CK biosynthesis increased along with a stimulation of CYP735A activity? For example, is the expression of IPT and CYP735A genes up-regulated in *cepr* mutants? What does it mean for the model proposed in Figure 5? If *cepr1* accumulates more tZ we may expect an inhibition of the primary root but this is counteracted by the mutation in the CEP receptor. CEPR is a negative regulator of tZ biosynthesis but CEPR and tZ are required for primary root growth inhibition etc....

(Additional remark: It is also surprising to observe the higher concentrations for cZ species because most of the time they are accumulated at a very low level?)

Additional comments:

All root growth data are presented through the relative root growth (%) and I can understand that this makes the results much more easier to display and interpret. However, I think it would be fair to present, in supplemental figures for example, the raw values for primary root growth because it is known that all these mutants have particular root phenotypes.

I am confused about the fact that the authors do not use the same post-hoc test following the ANOVA (Fisher's LSD versus Tukey HSD). Is there any explanation that may be added in the material and methods?

Please provide more details on Control Figure 2b (tZ is diluted in DMSO, even if the concentration is very low, the authors have to provide the exact control for these treatments).

Figure 2b, why the authors did not use the highest tZ concentration (100nM in Figure 3a)?

Recently, a paper has been published on the role of CK signaling in primary root growth in response to nitrate. In the context of this study, it may be worth to mention these results at least in the introduction (Naulin et al., 2020 Plant Cell Physiology: 61 p342).

Reviewer #2 (Remarks to the Author):

Plants rely on long-distance signaling to coordinate above- and below-ground growth in response to their environment. Cytokinin and CEP signaling pathways play a substantial role in this long-distance coordination. These pathways are currently assumed to function in parallel. In this manuscript, Taleski et al. provide genetic data indicating that these two pathways may intersect. The main basis for this argument relies on the fact that overexpression and knockout mutants in either the CEP or cytokinin signaling pathway show altered sensitivity, based on root length assays, to both CEP and cytokinin treatments. The authors go a few steps further, using hypocotyl grafting to dissect the extent to which key signaling events occur in the root or the shoot, and using cytokinin profiling to show that CEP pathway mutants have altered cytokinin levels.

The data presented in this manuscript constitutes a solid starting point to investigate the intersection of these two signaling pathways; however, almost all of the evidence supporting pathway intersection comes from a single type of assay looking at root length. Additional experiments that independently support the intersection of these pathways, and a deeper look into the molecular mechanism through which these pathways intersect is necessary to support the main conclusions of this study. Specifically, the authors provide no molecular detail supporting the interaction between cytokinin response and CEPD expression. Gene expression and protein abundance assays investigating the expression and mobility of CEPD in response to cytokinin and evidence for an interaction between AHK2/3 and CEPD would significantly substantiate

the model presented in this manuscript.

Specific comments:

In addition to the major comments above. We suggest including images of representative roots from the root assays to support figures 1-3, and publishing the raw root length data as a supplemental file. The figure legends for figures 1-4 are currently written in the style of a methods section rather than as descriptive legends. Furthermore, some of the methods are lacking in detail. Two specific examples include the description for calculating root length, which lacks key information about whether just primary roots were measured or root length refers to the length of all roots in the root system, and the description of CEP and tZ treatment on grafted plants, where it's unclear when the plants were transferred onto the treatments. Finally, while the manuscript is generally well-written, it is also a bit dense and technical. We recommend removing extraneous details that are not directly focused on the pathway interactions at hand.

Response to Reviewer Comments

REVIEWER COMMENTS

Reviewer #1 (Remarks to the Author):

In the present manuscript, Taleski and collaborators investigate possible interactions between CEP3, C-terminally encoded peptide 3, and cytokinins (CK), which are both known to be part of nutritional systemic signaling pathways. To do so, the authors used a combination of genetic, grafting and “pharmacological” (CEP3 or tZ external supply) approaches.

Results and discussion are combined and divided in 6 parts summarized below:

- 1) The role of CK in the CEP3-dependent inhibition of the primary root growth is assessed using a set of CK biosynthesis (*ipt3,5,7*, *cyp735a1,2*), transport (*abcg14*) and signaling mutants (*ahks*, type-B *arrs*). In *ipt3,5,7*, *cyp735a1,2*, *abcg14*, *ahk2,3* (mutant and gain-of-function) and *arr1,2,12* mutants, the inhibition of the primary root growth by CEP3 is altered (i.e., reduced in mutants).
- 2) Conversely, the inhibition of primary root growth by exogenous tZ application would require the CEP receptor as indicated by a less/absence of tZ-response in *cepr1* mutant.
- 3) Grafting experiments using a CEP3 overexpressor line and *ahk2,3* mutant show that i) CEP3 expression in roots has the greatest effect on primary root growth and ii) that this inhibition is reduced when AHK2 and 3 are mutated in the shoots
- 4) CK content analysis showed that the concentration of tZ-types is affected in roots and shoots by *cepr1* mutation (higher tZ-type content in root).
- 5) CEPD1 and 2, belonging to glutaredoxin family, are downstream CEP/CEPR to regulate NRT2.1 expression in response to N-limitation. Here, the authors showed that CEPD1 and 2 are required for CEP and tZ- inhibition of primary root growth.
- 6) This last paragraph proposes a model of interaction between CEP and tZ in response to environmental cues.

These results are original and interesting since i) CEP/CEPR/CEPD signaling pathway was known to regulate NRT2.1 and not known to regulate the primary root growth, although CEP3/CEPR1 was known to be involved (Delay et al. 2019), ii) the interaction between CEPDL (CEPD-like) and tZ has been shown but not between CEPD signaling pathway and tZ.

We thank the reviewer for these supportive comments.

To my opinion, these results are interesting but may be presented differently to highlight their novelty. The role of CEPD1&2 in CEP3-dependent primary root growth regulation has not been shown previously. However, this result is presented only in the middle of the graph in Figure 4. Why the authors did not show this result in a 1st figure?

The title of the paper is on the intersection of CEP and cytokinin pathways to inhibit root growth via CEPDs. So the intersection of the CEP and cytokinin pathways needs to be defined first, then the route through which it occurs. Both are novel, not just the route through CEPDs. Therefore, we feel that the flow of the manuscript, particularly with the addition of new data, is better served with this data remaining near the end of the manuscript. To strengthen the data on CEPDs and its role in CEP-dependent root growth inhibition, we have separated out the +CEP data originally in the middle of the graph in original Figure 4 as suggested, and grouped it with additional data relating to this topic in the new **Fig. 6**.

In addition, I think that the study would have a greater impact if the authors determine whether or not the control of the primary root growth by CEPDs depends on their activity in shoots or roots (grafting)? Indeed, previously the authors have shown that the control of the primary root growth by CEP3 depends on *cepr1* activity in roots and shoots. So, this experiment would clarify the role of this signaling pathway at a systemic level.

We agree with the reviewer. We present the requested grafting data testing whether shoot or root *CEPD1,2* activity is involved in CEP3-dependent primary root growth regulation in the new **Fig. 6c**. The results indicate that shoot CEPD activity contributes to CEP-dependent root growth inhibition, consistent with previously reported roles for CEPDs in N-demand signalling (Ohkubo et al 2017). This conclusion is reinforced by additional data obtained using a segmented agar plate setup, which we devised to address the reviewer's comments. The segmented-agar approach allows treatments to be applied specifically to roots or shoots to monitor their organ-specific effects (**Fig. 6d**; **Supplementary Fig. 9d**). This data shows that CEP3 treatment of shoots alone is sufficient for maximal primary root growth inhibition (**Supplementary Fig. 9e**), and that the *cepd1,2* mutant roots are less inhibited by shoot CEP3 treatment (**Fig. 6e**).

Figures 1 and 2 display very clear results on the role of CK in CEP3-mediated root growth inhibition and CEP/CEPR in tZ-mediated root growth inhibition. Therefore, I would suggest to add in Figure 2 the results presented in Figure 4 (only the +tZ treatment in *cepr1* and *cepd1/2* mutants).

We agree it is better to present the +CEP and +tZ data in the old figure 4 separately. To address this request, data showing +tZ treatment of *cepr1* and *cepd1/2* mutants now is grouped with additional thematically-related data in the new **Fig. 5**. So we have separated the data on the role of CEPDs in tZ mediated root growth inhibition (**Fig. 5**), from CEP mediated root growth inhibition (**Fig. 6**). So that the same control group data is not shown twice between the two figures, we used new data from a repeat of the experiment with congruent results for the tZ treatment in the new **Fig 5**.

In Figure 3, the role of *ahk2,3* in CEP3-mediated primary root growth inhibition seems to

depend on the activity of these receptors in shoots. However, I think that a last control is missing to reinforce the conclusion. Indeed, CEP3ox has the strongest phenotype when used as the rootstock but the role of *ahk2,3* in roots is tested only with CEP3ox shoots. Therefore, I would recommend to obtain a *ahk2,3* mutant overexpressing CEP3 and to use it as a rootstock (this control could even replace the last bar of the graph Figure 3b).

This is a good point from the reviewer. Indeed, the full extent of root AHK2,3 contribution to CEP sensitivity could not be assessed without the cross recommended. Thus, we obtained an *ahk2,3* mutant line overexpressing CEP3 via crossing as recommended. Reciprocal grafting between CEP3 overexpressing plants in the wild type and *ahk2,3* background suggest AHK2,3 in both the roots and shoots contributes to the inhibitory effect of CEP3 overexpression on root growth (new **Fig. 2e**). We believe that the new grafting data is more informative on the interaction between CEP3 and AHK2,3, and more straightforward to interpret than the original grafting data, so the new grafting data in updated **Fig. 2e** now replaces the old Fig. 3 grafting data.

To comfort the model, in particular box2, it will be worth to add a similar study using *arr1/2/12* mutants

Whilst this is an interesting question, the focus point for the cytokinin pathway in this manuscript is on the AHK2,3 cytokinin receptors. This connection to AHK2,3 has been expanded on in the revised manuscript and AHK2,3-associated data in the main paper is now grouped thematically in **Fig. 2** (the cytokinin biosynthesis and transport mutant data now appears separately in **Fig. 1**). As the connection to *arr1/2/12* is not the focus of the manuscript, we have removed *arr1/2/12* from the updated model and moved the *arr1/2/12* data from the main figures to **Supplementary Fig 5**. We think that grafting data with *arr1/2/12* is not critical to this manuscripts conclusions, and would be better suited to future studies further elucidating the intersection of CEP with cytokinin signalling downstream of AHK2,3 receptors.

(Additional remark: what is the rationale to focus only on these ARR's? I guess it's their link with WUSCHEL activation?).

We assessed these ARR's as they have known roles in cytokinin dependent root growth inhibition (Mason et al 2005. *Plant Cell*; Dello loio et al 2007. *Current Biology*).

The impact of this study will be also reinforced if the authors would add some grafting experiments combining *cepr1* mutant and +tZ treatments, in order to parallel the study with the CK receptor.

Since the point of convergence between CEP and cytokinin dependent inhibition of root growth involves CEPD, we have chosen a different approach. We assessed the contribution by the CEP pathway in root or shoots to cytokinin-dependent root growth inhibition using grafting with the *cepd1,2* double mutant (**Fig. 5c**). This Fig shows that CEPD1,2 activity in the

roots is required for tZ-dependent root growth inhibition. These conclusions are also supported by a segmented plate assay which shows that local tZ application to roots is required for root inhibition, and that *cepd1,2* is less sensitive to this treatment (**Fig 5d,e**). We don't find this too surprising since our published transcriptomic studies (Chapman et al 2019) indicate that *CEPD1* is downregulated more than 13-fold in the *cepr1* roots. Moreover, we show that CEP addition increases *CEPD1* expression in roots (**Fig. 4f**) as does tZ treatment (**Fig. 4b**), and *CEPD1* expression is basally lower in *ahk2,3* mutant roots (**Fig. 4f**). This unequivocally shows, in addition to systemic CEPD circuits, there is a local CEP/cytokinin dependent control of *CEPD1* expression in roots. To reflect these findings, "Systemic" has been removed from the paper's title to also be inclusive of local signalling.

In Table 1, the authors showed that the accumulation of tZ is altered in *cepr1* mutants, which is a very interesting result as well. Unfortunately, this result is only described in the corresponding paragraph but not really interpreted and not even display in the model (Figure 5). The results show most of the time that tZ concentration is higher in mutants whereas all the other concentrations are not modified. What does it mean? Is the CK biosynthesis increased along with a stimulation of CYP735A activity? For example, is the expression of IPT and CYP735A genes up-regulated in *cepr* mutants? What does it mean for the model proposed in Figure 5? If *cepr1* accumulates more tZ we may expect an inhibition of the primary root but this is counteracted by the mutation in the CEP receptor. CEPR is a negative regulator of tZ biosynthesis but CEPR and tZ are required for primary root growth inhibition etc.... (Additional remark: It is also surprising to observe the higher concentrations for cZ species because most of the time they are accumulated at a very low level?)

The increased tZ-type cytokinins in *cepr1* roots is consistent with feedback-mediated upregulation of tZ biosynthesis. We have expanded on this interpretation in the manuscript (**lines 171-173**). Supporting this interpretation of feedback upregulation is that the *ahk2,3* mutant, which has reduced tZ sensitivity, also has elevated tZ (Riefler et al 2006). As neither *cepr1* or *ahk2,3* have short roots despite having elevated tZ, clearly the sensitivity to tZ is reduced in both lines. Reduced tZ sensitivity in *cepr1* is consistent with our root growth data presented for *cepr1* v wild type in response to increasing tZ levels (**Fig 3a,b**).

We have published transcriptomic data for *cepr1* roots (Chapman et al 2019) which shows that the main CYP735A gene contributing to tZ biosynthesis (*CYP735A2*) is in fact downregulated in *cepr1*, suggesting the transcript levels of this gene are not necessarily correlated with tZ levels. CYP735A1 and IPT genes are not differentially regulated in *cepr1* in this dataset, apart from an upregulation of *IPT7*. How compensatory feedback mechanisms work to determine cytokinin pools in the absence of functional CEPR1 may be complicated and is beyond the core focus of this paper. Therefore, for simplicity, we have not included

feedback control in the model figure, however we have touched on it in a relevant section of the discussion text (see lines 227-229).

With regards to the observed cZ levels, it is possible for cZ-types to make up a relatively large proportion of total cytokinin types in samples from Arabidopsis (see for example Poitout et al 2018. *Plant Cell*). We speculate that cZ-type levels will vary depending on factors such as the growth conditions, plant age, genetic background, and tissues harvested.

Additional comments:

All root growth data are presented through the relative root growth (%) and I can understand that this makes the results much more easier to display and interpret. However, I think it would be fair to present, in supplemental figures for example, the raw values for primary root growth because it is known that all these mutants have particular root phenotypes.

We agree. As requested the raw values for primary root growth, when not appearing in main figures, are now presented in Supplementary figures.

I am confused about the fact that the authors do not use the same post-hoc test following the ANOVA (Fisher's LSD versus Tukey HSD). Is there any explanation that may be added in the material and methods?

Tukey HSD test are now used throughout the manuscript where multiple pairwise comparisons are made for consistency.

Please provide more details on Control Figure 2b (tZ is diluted in DMSO, even if the concentration is very low, the authors have to provide the exact control for these treatments).

The control for this experiment does not use DMSO, however the concentration used in the tZ treatment is very low (0.001% v/v) and does not influence primary root growth, as shown in the data presented below for the reviewers.

DMSO content in tZ treatment does not impact root growth. Primary root growth of Col-0 plants grown for 12 days on ½ MS medium without added DMSO (-DMSO), or with DMSO (0.001% v/v). Not significantly different; Two sample T-test ($p > 0.05$). $n = 16-17$.

Figure 2b, why the authors did not use the highest tZ concentration (100nM in Figure 3a)?

10nM rather than 100nM was used as at 10nM there was a differential presence/absence of root growth inhibition between WT and *cepr1* at 10nM, and thus differentiates the two lines more clearly.

Recently, a paper has been published on the role of CK signaling in primary root growth in response to nitrate. In the context of this study, it may be worth to mention these results at least in the introduction (Naulin et al., 2020 Plant Cell Physiology: 61 p342).

This reference has been added to a relevant section of the intro (**line 67**).

Reviewer #2 (Remarks to the Author):

Plants rely on long-distance signaling to coordinate above- and below-ground growth in response to their environment. Cytokinin and CEP signaling pathways play a substantial role in this long-distance coordination. These pathways are currently assumed to function in parallel. In this manuscript, Taleski et al. provide genetic data indicating that these two pathways may intersect. The main basis for this argument relies on the fact that overexpression and knockout mutants in either the CEP or cytokinin signaling pathway show altered sensitivity, based on root length assays, to both CEP and cytokinin treatments. The authors go a few steps further, using hypocotyl grafting to dissect the extent to which key signaling events occur in the root or the shoot, and using cytokinin profiling to show that CEP pathway mutants have altered cytokinin levels.

The data presented in this manuscript constitutes a solid starting point to investigate the intersection of these two signaling pathways; however, almost all of the evidence supporting pathway intersection comes from a single type of assay looking at root length. Additional experiments that independently support the intersection of these pathways, and a deeper look into the molecular mechanism through which these pathways intersect is necessary to support the main conclusions of this study. Specifically, the authors provide no molecular detail supporting the interaction between cytokinin response and CEPD expression. Gene expression and protein abundance assays investigating the expression and mobility of CEPD in response to cytokinin and evidence for an interaction between AHK2/3 and CEPD would significantly substantiate the model presented in this manuscript.

We thank the reviewer for their suggestions, and agree that the addition of molecular data would reinforce the conclusions of the paper. We have carried out a number of additional experiments to address this which are featured in the new **Fig. 4**. This includes a qRT-PCR experiment assessing *CEPD* expression in response to tZ treatment (along with *ARR5* as a tZ-responsive marker gene) (**Fig. 4a-c**). This shows that *CEPD1* expression is upregulated in roots by tZ treatment, which supports the interaction between the cytokinin response and *CEPD* expression. We also show via qRT-PCR that *CEPD1* expression is basally lower in the *ahk2,3* mutant roots compared to wild type (**Fig. 4f**), which is molecular evidence for an interaction between AHK2/3 and CEPD. We also assessed a GFP-CEPD1 reporter (Ohkubo et al 2017) in response to tZ treatment (as well as to CEP, as a positive control) (**Fig. 4d,e**). GFP-CEPD1 fluorescence was increased in the primary root in response to tZ (in addition to the expected increase in response to CEP). This result is consistent with increased CEPD1 protein levels in the primary roots in response to cytokinin. With respect to mobility of CEPD in response to cytokinin, it can be inferred from reciprocal grafting between wild type and *cepd1,2* in combination with tZ treatment (**Fig 5c**) that CEPDs are produced locally in the root in response to cytokinin, which is congruent with our qRT-PCR data showing *CEPD1* is upregulated in roots (**Fig 4b**), and segmented agar plate assays showing *cepd1,2* primary roots are less inhibited by local tZ application to roots (**Fig. 5d,e**). This is an interesting distinction to the long-distance CEP-dependent root growth inhibition, which we show can be elicited via application of CEP to the shoot alone (**Fig. 6d,e; Supplementary Fig. 9d,e**), and depends on *CEPD1,2* gene function in the shoot as assessed via grafting (**Fig. 6c**). Given the CEP and cytokinin pathways converge on *CEPD* activity in different organs, we have updated the abstract (lines **36-38**) and model (**fig. 7**) along with associated main text to reflect this finding.

Furthermore, to supplement our original root length data using a different assay type, we have added data assessing primary root meristem cell numbers (**Supplementary Figures 3,4,8**), as both CEP (Delay et al 2019) and tZ (Dello Ioio et al 2007) are known to affect this. These data suggest that CEP inhibition of meristem cell number involves AHK2,3 activity (**Supplementary Figures 3,4**), and tZ inhibition of meristem cell number involves CEPD1,2 activity (**Supplementary Figures 8**). We think these new data bolster our original conclusions and gives further insight into how CEP and cytokinin pathways impact primary root growth.

Specific comments:

In addition to the major comments above. We suggest including images of representative roots from the root assays to support figures 1-3, and publishing the raw root length data as a supplemental file.

Representative roots from the assays as well as the raw root length data now appear either in the supplementary data or in certain instances in the updated main figures.

The figure legends for figures 1-4 are currently written in the style of a methods section rather than as descriptive legends.

We have updated the figure legends to reduce the level of detail more suited to the methods section. Where it doesn't impact comprehension of the figure, these specific details have been moved to the methods section.

Furthermore, some of the methods are lacking in detail. Two specific examples include the description for calculating root length, which lacks key information about whether just primary roots were measured or root length refers to the length of all roots in the root system, and the description of CEP and tZ treatment on grafted plants, where it's unclear when the plants were transferred onto the treatments.

Primary roots were measured, which is now specified in the methods (**line 257**). The updated methods section also details specifically when grafted plants were transferred to CEP or tZ treatments (see Grafting section **lines 264-273**).

Finally, while the manuscript is generally well-written, it is also a bit dense and technical. We recommend removing extraneous details that are not directly focused on the pathway interactions at hand.

Where appropriate we have removed extraneous and technical detail in the revised manuscript. In particular, we have reduced the complexity of the model (new **Fig. 7**) to reflect only the core signalling components focussed on in the manuscript.

Reviewer #1 (Remarks to the Author):

Dear Authors,

In the present manuscript, you investigate the interaction between CEPs peptides and cytokinins which are both known to be part of signalling pathways controlling root development. In this revised manuscript, you addressed a lot of points raised by the 2 reviewers. In your rebuttal, you gave good arguments to explain some of your choices.

This study is interesting since it explores the connection between CK and CEP-signaling pathways to control primary root growth. CK and CEPD-L connection has been proposed before to control nitrate transport (Ota et al. 2020)

The results that you showed suggested that indeed these 2 pathways are not independent. However, I don't think that your results show that the convergence point is CEPD1,2. All experiments show reciprocal influence but do not explain mechanistically how CK and CEP-CEPD pathways interact. For instance, I am questioning the results that you obtained in figure 5: 1) Fig 5.b, the insensitivity of *cepr1* mutant is no longer observed (a little bit reduced according to the statistics); 2) Supp Fig7.c, the homo-grafting *cepd1,2/cep1,2* does not display any significant increase of primary root growth meaning that already the control is not good and 3) I am totally lost with the results in Fig5.e : you conclude that tZ inhibits root growth via CEPDs in roots and not in shoots. However, tZ application on shoots (apparently you tested a unique concentration) does not have any effect on primary root growth. If tZ application in these conditions has no effect how one could conclude that CEPD is required or not?

I have additional comments:

A lot of your experiments rely on the application and the effect of a 15 amino acid peptide, CEP3. You never tested or showed any experiments ruling out that the effect you observed it's not related to the provision of any form of organic nitrogen. Even with the CEP3ox line, we could ask if the overexpression of any other 15aa peptide would have a similar effect. Organic nitrogen as amino acids and peptides are more and more studied for their effect on plant growth and yield and. One could ask the level of specificity of what is observed here.

You generated the line *ahk2,3/CEP3ox* and did some grafting experiments suggesting that CEP3-related signaling rely on CK signaling in roots. The homo-grafting *ahk2,3/ahk2,3* as a control is missing.

In Figure 3c, you showed that *cepr1* is insensitive to CEP3 and tZ. Is this mutant insensitive to any treatments? A positive effect of a root treatment by other chemicals or hormones would be certainly a good control to ensure that it's not just a structural feature of this mutant.

The number of plants displayed on every bar graphs is rather low (between 5 and 18). For primary root measurements, one would expect a greater number of plants since it's an easy measurement. I do not see the extra-value of displaying the relative root growth compared to what is presented in supplemental. It's rather misleading about the root growth of all genotypes in basal conditions.

Reviewer #2 (Remarks to the Author):

The revised manuscript, "CEP peptide and cytokinin pathways converge on CEPD glutaredoxins to inhibit root growth" is much improved and satisfies my original comments on the first submission. Figure 4 provides substantial molecular detail regarding the interaction between cytokinin and CEPD, and the root versus shoot intersection of cytokinin versus CEP regulation of root architecture. I have two recommendations that would improve the thoroughness and quality of the present study:

First, for figure 4, some additional expression measurements would improve the study. For example, in figure 4 D,E, the authors should include a similar fluorescent measurement in shoot vasculature. According to the model (presented in figure 7), CEPD should be upregulated in shoot vasculature and mobilized to the root in response to CEP3, while it should just be upregulated in

root vasculature in response to cytokinin. The qRT-PCR data presented in Figure 4F is not perfectly congruent with this model, as the authors show that CEPD1 is upregulated in roots in response to CEP3 application, and they don't provide data for shoot-expressed CEPD1 in response to CEP3 application. One more qRT-PCR experiment measuring CEPD expression in the shoot in response to CEP3 application, and fluorescent measurements for CEPD1 expression in shoot vasculature in response to CEP3 application versus cytokinin application would complete this model.

My second recommendation is easy to correct. All of the bar chart data would be better represented in boxplot form. Boxplots show data distribution more clearly and make it easier for the reader to compare distributions across samples.

RESPONSE TO REVIEWER COMMENTS

REVIEWER COMMENTS

Reviewer #1 (Remarks to the Author):

Dear Authors,

In the present manuscript, you investigate the interaction between CEPs peptides and cytokinins which are both known to be part of signalling pathways controlling root development.

In this revised manuscript, you addressed a lot of points raised by the 2 reviewers. In your rebuttal, you gave good arguments to explain some of your choices.

This study is interesting since it explores the connection between CK and CEP-signaling pathways to control primary root growth. CK and CEPD-L connection has been proposed before to control nitrate transport (Ota et al. 2020)

The results that you showed suggested that indeed these 2 pathways are not independent. However, I don't think that your results show that the convergence point is CEPD1,2. All experiments show reciprocal influence but do not explain mechanistically how CK and CEP-CEPD pathways interact. For instance, I am questioning the results that you obtained in figure 5:

1) Fig 5.b, the insensitivity of *cepr1* mutant is no longer observed (a little bit reduced according to the statistics);

The insensitivity at 10nM tZ was in *cepr1-3* in the Col-0 background (Fig 3) whereas Fig5b is *cepr1-1* in the No-0 background as in this case we are comparing to No-0 *cepd1,2* (Ohkubo et al 2017). The *cepr1* alleles and WT backgrounds used are described in the figure legends and methods. We find that the No-0 background is more sensitive to tZ than the Col-0 background, which can explain why *cepr1-1* is partially insensitive rather than fully insensitive to 10nM tZ in Fig5b. Different ecotype backgrounds are known to vary in their magnitude of response to hormones including cytokinin (Ristova et al 2018 *Plant Journal*). We find it reassuring that the direction of the effect is the same in independent *cepr1* alleles in different ecotype backgrounds (i.e. less sensitivity to tZ in *cepr1* compared to WT).

2) Supp Fig7.c, the homo-grafting *cepd1,2/cepd1,2* does not display any significant increase of primary root growth meaning that already the control is not good

Grafting does lead to a certain amount of variation, and whilst not statistically significant using a Tukey HSD test (a fairly conservative test), the tendency for *cepd1,2/cepd1,2* having longer primary roots than WT/WT is still observed. If one were to just consider those two groups (*cepd1,2/cepd1,2* control v.s. WT/WT control) and perform a two sample t-test, *cepd1,2/cepd1,2* is significantly longer than WT/WT

($p= 0.0187$). More importantly, *cepd1,2/cepd1,2* is clearly less sensitive to tZ than WT/WT, and therefore the homograft controls are valid.

and 3) I am totally lost with the results in Fig5.e : you conclude that tZ inhibits root growth via CEPDs in roots and not in shoots. However, tZ application on shoots (apparently you tested a unique concentration) does not have any effect on primary root growth. If tZ application in these conditions has no effect how one could conclude that CEPD is required or not?

It is not well understood where cytokinins act (in root or shoot) to inhibit primary root growth. Fig5e now appears as Fig5d, as we have combined the data in Fig5e with the diagrammatic representation of the experimental set-up in Fig5d to improve clarity. The experiment in Fig5d is devised to i) test if root and/or shoot application of tZ is sufficient to inhibit root growth, ii) test if this inhibition depends on CEPDs. We know that 10nM tZ addition inhibits root growth (Fig. 5d uses 10nM tZ, which is used elsewhere e.g. fig5b, so this is not a unique concentration). If tZ acts via the shoot, then you should be able to bypass the necessity to apply to the root (see for example the activity of CEP3 when applied to shoot only in Fig 6d). Indeed 10nM tZ application to shoots alone does not inhibit root growth, we are not claiming it does nor claiming a role for CEPD in this. Instead application of 10nM tZ to roots only was necessary and sufficient for root growth inhibition (it inhibits to the same extent as if it were applied to both the root and shoot, see the two grouped columns on the right side of Fig5d). Moreover, *cepd1,2* was less sensitive to inhibition of root growth by tZ treatment to roots (column on furthest right side of graph). This supports local activity of tZ in root involving CEPD function in the inhibition of primary root growth, which is also supported by grafting experiment in Fig.5c, and the gene expression data Fig 4.

I have additional comments:

A lot of your experiments rely on the application and the effect of a 15 amino acid peptide, CEP3. You never tested or showed any experiments ruling out that the effect you observed it's not related to the provision of any form of organic nitrogen. Even with the CEP3ox line, we could ask if the overexpression of any other 15aa peptide would have a similar effect. Organic nitrogen as amino acids and peptides are more and more studied for their effect on plant growth and yield and. One could ask the level of specificity of what is observed here.

It has been demonstrated in Delay et al., 2013 that application of a "scrambled" CEP peptide with the same amino acids in different order does not have the inhibitory effect on root growth as CEP peptides. CEP3 has biological activity in the mid nM range (10^{-8} M; Delay et al 2013) which is low for it to be a nutritive effect of organic N provision. Most importantly, CEP peptides bind specifically to the ectodomain of the LRR-RLK receptor CEPR1 (Tabata et al 2014), and their biological activity is abolished in the *cepr1* knockout mutants (Tabata et al 2014, Ohkubo et al 2017, Chapman et al 2019, Delay et al 2019, Chapman et al 2020). Therefore, the activity of CEP3 is consistent with being a peptide hormone acting specifically through CEPR1, rather than having its effects through organic N provision.

You generated the line *ahk2,3/CEP3ox* and did some grafting experiments suggesting that CEP3-related signaling rely on CK signaling in roots. The homo-grafting *ahk2,3/ahk2,3* as a control is missing.

We decided to not include the *ahk2,3/ahk2,3* graft and to increase the number of plants for the other grafting combinations. This graft is not required for investigating whether CK perception in the root or shoot is required for CEP perception and the correct controls are the CEPox homograft as well as the CEPox *ahk2,3* homograft as these are the lines which are actually grafted with each other.

In Figure 3c, you showed that *cepr1* is insensitive to CEP3 and tZ. Is this mutant insensitive to any treatments? A positive effect of a root treatment by other chemicals or hormones would be certainly a good control to ensure that it's not just a structural feature of this mutant.

We present data below for reviewer #1 showing the effect of 100nM 1-Naphthaleneacetic acid (NAA) on primary root growth in Col-0 and *cepr1-3*. This shows *cepr1* is sensitive to this treatment, and has the same degree of sensitivity as Col-0. Thus, *cepr1* is clearly not insensitive to any treatments.

NAA treatment inhibits Col-0 and *cepr1-3* primary root growth. Plants were grown for 10 days on medium with or without 100nM NAA. **a** Absolute primary root length and, **b** relative primary root length expressed as a percentage of the respective control group for each line (n=7). Letters indicate significant differences (ANOVA followed by Tukey HSD test, $p < 0.05$). Box plot centre line, median; box limits, upper and lower quartiles; whiskers, 1.5x interquartile range.

The number of plants displayed on every bar graphs is rather low (between 5 and 18). For primary root measurements, one would expect a greater number of plants since it's an easy measurement.

I do not see the extra-value of displaying the relative root growth compared to what is presented in supplemental. It's rather misleading about the root growth of all genotypes in basal conditions.

As key root growth experiments have been repeated multiple times and statistical analyses have been performed, we are confident that our results are valid even though the number of plants might be low

in your opinion. In publication like e.g. Delay et al., 2013, Fonouni-Farde et al., 2019 or Lin et al., 2022 a similar number of plants/roots like in our experiments have been used.

With regard to presenting the relative root growth, the data is presented in this way so that differences in sensitivity to hormone treatments are easily interpreted and not obscured by the different genotypes having different basal primary root growth. Reviewer #1 acknowledged as such in the first round of review:

“All root growth data are presented through the relative root growth (%) and I can understand that this makes the results much more easier to display and interpret. However, I think it would be fair to present, in supplemental figures for example, the raw values for primary root growth because it is known that all these mutants have particular root phenotypes.”

We do not see how presenting the data in this way is misleading given the above quote, particular when we have presented the raw root growth data in the supps as was recommended by reviewer #1. Again, the key response we are looking at is sensitivity in the mutant backgrounds to CEP or tZ, not the basal root growth for mutants that have largely been characterised elsewhere.

Reviewer #2 (Remarks to the Author):

The revised manuscript, "CEP peptide and cytokinin pathways converge on CEPD glutaredoxins to inhibit root growth" is much improved and satisfies my original comments on the first submission. Figure 4 provides substantial molecular detail regarding the interaction between cytokinin and CEPD, and the root versus shoot intersection of cytokinin versus CEP regulation of root architecture. I have two recommendations that would improve the thoroughness and quality of the present study:

First, for figure 4, some additional expression measurements would improve the study. For example, in figure 4 D,E, the authors should include a similar fluorescent measurement in shoot vasculature. According to the model (presented in figure 7), CEPD should be upregulated in shoot vasculature and mobilized to the root in response to CEP3, while it should just be upregulated in root vasculature in response to cytokinin. The qRT-PCR data presented in Figure 4F is not perfectly congruent with this model, as the authors show that CEPD1 is upregulated in roots in response to CEP3 application, and they don't provide data for shoot-expressed CEPD1 in response to CEP3 application. One more qRT-PCR experiment measuring CEPD expression in the shoot in response to CEP3 application, and fluorescent measurements for CEPD1 expression in shoot vasculature in response to CEP3 application versus cytokinin application would complete this model.

We thank reviewer #2 for their supportive comments and helpful suggestions. We agree the manuscript is much improved, and that the suggested experiments would further reinforce the model. We carried out the two proposed experiments, which have been added to Fig 4. Fig 4g shows the requested qRT-

PCR experiment measuring CEPD expression in the shoot in response to CEP3 application. *CEPD1* and *CEPD2* expression in shoots was increased in response to CEP3 treatment, as expected. Fig 4h,i shows the requested fluorescent measurements for *CEPD1* expression in shoot vasculature in response to CEP3 application versus cytokinin application. This shows CEP3 treatment, but not tZ treatment induces GFP-CEPD1 fluorescence in the cotyledon vasculature. These new data support the expectation that CEPD is upregulated in shoot vasculature and mobilised to the root in response to CEP3, while cytokinin upregulates CEPD in the root vasculature. We have adjusted our model (fig 7) to acknowledge that CEP3 induces both *CEPD1* and *CEPD2*, whereas tZ induces only *CEPD1* transcriptionally but could possibly signal through CEPD2 via other mechanisms (e.g. post-translationally). We also thought it fair to acknowledge in the discussion that given the No-0 *cepd1,2* mutant is partially, but not fully insensitive to CEP or tZ, that other molecular components also contribute to CEP and tZ signaling and that future studies may extend analyses to *cepd* mutants in other *Arabidopsis* ecotypes and different plant species (lines 238-244). We think that these adjustments reflect our current understanding more precisely.

My second recommendation is easy to correct. All of the bar chart data would be better represented in boxplot form. Boxplots show data distribution more clearly and make it easier for the reader to compare distributions across samples.

Bar chart data is now presented in boxplot form, as requested.

Reviewer #1 (Remarks to the Author):

Dear Authors,

thanks for all your answers. I am satisfied with the new version of you paper.

Best wishes

Reviewer #2 (Remarks to the Author):

I am satisfied with the authors' responses to reviewer comments and updated figures. I have a minor recommendation to remove the word "specific" from line 138, revised sentence would read: "mutants support a role for AHK2 and AHK3 in CEP-mediated..." In the absence of a direct mechanism linking CEP and cytokinin signaling, I don't think the authors can claim a specific role for AHK2/AHK3 in mediating this genetic interaction.

RESPONSE TO REVIEWER COMMENTS

REVIEWERS' COMMENTS

Reviewer #1 (Remarks to the Author):

Dear Authors,

thanks for all your answers. I am satisfied with the new version of you paper.

Best wishes

We thank Reviewer #1 for their comments.

Reviewer #2 (Remarks to the Author):

I am satisfied with the authors' responses to reviewer comments and updated figures. I have a minor recommendation to remove the word "specific" from line 138, revised sentence would read: "mutants support a role for AHK2 and AHK3 in CEP-mediated..." In the absence of a direct mechanism linking CEP and cytokinin signaling, I don't think the authors can claim a specific role for AHK2/AHK3 in mediating this genetic interaction.

We thank Reviewer #2 for their comments. "Specific" has been removed from line 138 as recommended.